# Biosynthesis of Poly-(3-hydroxybutyrate) under the Control of an Anaerobically Induced Promoter by Recombinant *Escherichia coli* from Sucrose

**DOI:** 10.3390/molecules27010294

**Published:** 2022-01-04

**Authors:** Fangting Wu, Ying Zhou, Wenyu Pei, Yuhan Jiang, Xiaohui Yan, Hong Wu

**Affiliations:** 1Tianjin Key Laboratory of Conservation and Utilization of Animal Diversity, College of Life Sciences, Tianjin Normal University, Tianjin 300387, China; wft2021@163.com (F.W.); pwy13132023220@163.com (W.P.); jyhleo@163.com (Y.J.); 2School of Chinese Materia Medica, Tianjin University of Traditional Chinese Medicine, Tianjin 301617, China; zhouyingbiopharm@163.com; 3State Key Laboratory of Component-Based Chinese Medicine, Tianjin University of Traditional Chinese Medicine, Tianjin 301617, China

**Keywords:** polyhydroxyalkanoates, PHB, *Escherichia coli*, synthetic biology, anaerobic promoter

## Abstract

Poly-(3-hydroxybutyrate) (PHB) is a polyester with biodegradable and biocompatible characteristics and has many potential applications. To reduce the raw material costs and microbial energy consumption during PHB production, cheaper carbon sources such as sucrose were evaluated for the synthesis of PHB under anaerobic conditions. In this study, metabolic network analysis was conducted to construct an optimized pathway for PHB production using sucrose as the sole carbon source and to guide the gene knockout to reduce the generation of mixed acid byproducts. The plasmid pMCS-*sacC* was constructed to utilize sucrose as a sole carbon source, and the cascaded promoter P*_3nirB_* was used to enhance PHB synthesis under anaerobic conditions. The mixed acid fermentation pathway was knocked out in *Escherichia coli* S17-1 to reduce the synthesis of byproducts. As a result, PHB yield was improved to 80% in 6.21 g/L cell dry weight by the resulted recombinant *Escherichia coli* in a 5 L bed fermentation, using sucrose as the sole carbon source under anaerobic conditions. As a result, the production costs of PHB will be significantly reduced.

## 1. Introduction

Global warming concerns and environmental problems, such as microplastics and non-degradable plastics, are necessitating the transition from petrochemical-dependent industries into biorefineries and the production of polymers from renewable biomass-based resources such as microbes [1,2]. Researchers have emphasized the need to produce biodegradable bio-based plastics including polyhydroxyalkanoates (PHAs), polylatic acid (PLA), poly(butylene adipate-co-terephthalate) (PBAT), and so on, as a sustainable solution to the issues of plastic pollution and fuel depletion [3,4,5,6].

PHAs, as a family of biodegradable and biocompatible thermal polyesters, have attracted considerable attention to be developed as environmentally friendly bioplastics. The properties of PHAs are similar to those of conventional plastics [7]. Poly(3-hydroxybutyrate) (PHB) is the most common and well-studied member among the PHAs [8,9], and the favorable properties of PHB have led to its development and utilization as a biodegradable material in many fields such as wrapping films, bags, and bottles [10,11]. PHB can accumulate in various microorganisms from the natural environment, including *Alcaligenes*, *Bacillus*, and *Pseudomonas* [12,13]. Benefiting from the well-characterized genome and the wide availability of tools for genetic manipulation, *Escherichia coli* expressing acetyl-CoA acetyltransferase (PhaA), acetoacetyl-CoA reductase (PhaB), and PHA synthase (PhaC) has been extensively studied for PHB production, particularly for high-cell-density cultivation in bioreactors [14,15,16,17]. Process optimization and metabolic engineering strategies have facilitated a deeper understanding of PHB production by recombinant *E. coli* from renewable carbon sources. However, the high costs of fermentation substrates and the remarkable energy consumption during the fermentation processes had impeded the application of PHB as a bio-based plastic [18,19].

Synthesis of PHB via recombinant *E. coli* strain has received extensive attention, and numerous studies attempted to reduce the production cost of PHB to promote its commercialization [20,21,22,23,24]. For example, by optimizing the metabolic pathways of PHB, an engineered *E. coli* can use crude glycerol as the sole carbon source to produce PHB and reach a PHB content of 65% dry cell weight after fermentation [20]. Moreover, construction of the sucrose utilization pathway in an engineered *E. coli* XL1-Blue strain resulted in the production of 38 wt.% PHB with 20 g/L sucrose as the sole carbon source [21]. In addition to expanding the repertoire of the widely available and low-cost substrates such as sucrose, xylose, and glycerol, many studies also aimed at reducing the energy consumption in the fermentation processes, by utilizing hypoxic conditions that are more conducive to the accumulation of PHB [22,23,24].

Current research interest in PHB production mainly focuses on the application of metabolic engineering and synthetic biology strategies to increase PHB contents at a relatively low cost [25,26]. If the host strain can utilize the cheap and abundant carbon sources, the cost of PHB production will be significantly reduced, which is eventually beneficial for the popularization of PHB. Sucrose is an inexpensive and readily available substrate in sugarcane molasses and can be used as an alternative carbon source to glucose in *E. coli* [27,28]. The ability of sucrose utilization in an *E. coli* strain depends on the β-fructofuranosidase system, which is secreted into the culture medium where it can break down sucrose into glucose and fructose. When an *E. coli* strain did not have such enzymes, it could not use sucrose as carbon source. In order to produce PHB from sucrose in *E. coli* strains lacking *sacC*, both sucrose utilization and PHB biosynthesis pathways needed to be established. A sucrose utilization pathway was constructed in *E. coli* by expressing the β-fructofuranosidase (SacC) from *Mannheimia succiniciproducens* MBEL 55E [29]. By introducing the *sacC* gene, *E. coli* could convert sucrose into glucose and fructose [21]. In this way, the recombinant strain becomes a cost-effective and sustainable host to produce PHB and other materials.

The minimal cell growth rate, when the dissolved oxygen level cannot match the demand of cell growth, can be resolved with the inclusion of air compressors and agitation, but these efforts incur high energy consumption and additional costs during the fermentation process. *E. coli* can grow under microaerobic or anaerobic conditions; therefore, it is important to find efficient promoters that could respond to oxygen limitation to control gene expression under anaerobic conditions [30]. Anaerobic promoters that are inducible under a low dissolved oxygen level, such as P*_adhE_* and P*_vgb_*, could enhance PHB accumulation [31,32]. Indeed, PHB accumulation was increased by 18% under the control of the alcohol dehydrogenase promoter, P*_adhE_* [30]. The cascaded *vgb* promoter, P*_8vgb_*, improved PHB yield to more than 90% in recombinant *E. coli* under the low-aeration condition [33]. Furthermore, the bacterial cells are subjected to a low dissolved oxygen level and acidic byproducts. To overcome these challenges, *E. coli* strains with knocked out genes related to acid synthesis (for example, *ackA-pta*, *poxB*, and *ldhA*) have already been developed as a suitable host for PHB production under microaerobic conditions [34].

In this study, three *E. coli* strains (S17-1, DH5α, and Top10) were engineered to utilize sucrose as the sole carbon source using a β-galactosidase-expressing plasmid. Furthermore, the anaerobic promoter P*_nirB_*, which controls the expression of the nitrite reductase in *E. coli*, was selected to control the expression of the *phaCAB* operon in *E. coli* [35]. Their abilities to produce PHB under microaerobic conditions were compared. The genes involved in acid formation in *E. coli* were then knocked out in the most effective strain, and the anaerobic promoter P*_nirB_* was cascaded to enhance PHB production under microaerobic conditions. Findings from this study offer a deeper understanding of PHB production in *E. coli* with anaerobic promoter expression systems.

## 2. Results and Discussion

### 2.1. Metabolic Engineering Strategies for PHB Production from Sucrose

The β-fructofuranosidase can break down sucrose into glucose and fructose. By engineering the pathway of the β-fructofuranosidase system, recombinant *E. coli* and *Ralstonia eutropha* have already been successfully modified to synthesize PHB with sucrose as the sole carbon source [21,36]. In this study, the β-fructofuranosidase-encoding gene was expressed in three different *E. coli* strains: S17-1, DH5α, and Top10, to compare their ability to produce PHB from sucrose in flask culture (20 g/L sucrose). Three mutant strains were all established by introducing the pBHR68 plasmid and the pMCS-sacC plasmid into these three *E. coli* strains. The pBHR68 plasmid contains the phaCAB genes from *R. eutropha* and the pMCS-sacC plasmid harbors the β-fructofuranosidase-encoding gene (*sacC*) from *M. succiniciproducens* (Figure 1).

The concentrations of sucrose, glucose, and fructose were measured to determine sucrose utilization by the engineered pathway with the *sacC* gene in the flask culture. Sucrose was rapidly hydrolyzed into glucose and fructose initially, indicating that the sucrose utilization pathway had been successfully established in the host strains. The concentrations of remnant glucose were 3.40 ± 0.39 g/L (S17-1), 3.63 ± 0.23 g/L (DH5α), and 3.33± 0.48 g/L (Top10), while the concentrations of remnant fructose were 8.62 ± 0.68 g/L (S17-1), 8.55 ± 0.57 g/L (DH5α), and 8.79 ± 0.36 g/L (Top10) (Figure 2). The remnant fructose concentrations were relatively lower than that of glucose in all the three strains after cultivation. Thus, by expressing the *sacC* gene to hydrolyze sucrose, fructose was not utilized as efficiently as glucose in the recombinant *E. coli* strains (Figure 2A). This was consistent with the trend of sucrose utilization for the production of PHAs in a previous study [21]. During cultivation of the recombinant strain using sucrose, glucose was almost entirely utilized while fructose was not fully consumed.

To identify differences in PHB accumulation by the host strains, the CDW (cell dry weight) and intracellular PHB content were measured after cultivation for 48 h. *E. coli* S17-1, DH5α, and TOP10 harboring plasmids pBHR68 and pMCS-sacC were cultivated in minimal medium (MM medium) with added 20 g/L sucrose. Final CDW values were 1.51 ± 0.52 g/L (S17-1), 1.02 ± 0.27 g/L (DH5α), and 0.50 ± 0.02 g/L (Top10). All the three recombinant *E. coli* strains could accumulate more than 35% PHB. The final PHB contents (wt.%) were 56% (S17-1), 53% (DH5α), and 38% (Top10). Among them, the highest PHB content was achieved by the recombinant S17-1 strain, while the lowest content was observed in Top10 (Figure 2B). S17-1 (pBHR68 and pMCS-sacC) yielded a maximal PHB content of 0.85 g/L and was the most suitable host for PHB accumulation using sucrose as substrate. Utilization of sucrose by the recombinant strains could facilitate low-cost PHB production, which will increase economic competitiveness.

### 2.2. Genetically Modified Promoter P_nirB_ under Anaerobic Condition

Microaerobic and anaerobic conditions were reported to reduce the costs for microbial production of PHB [37]. P_nirB_ is the promoter controlling the expression of the first gene of the *nir* operon in *E. coli* induced in anaerobic conditions. In order to achieve high PHB content under microaerobic conditions in *E. coli*, the *nirB* promoter was used to control the PHB biosynthesis operon *pha*CAB.

Assembly of repeated P_vgb_ promoters was previously shown to create the strongest induction [33]; thus, this tandem repeat method was employed with the P_nirB_ promoter to improve cell growth under oxygen-limitation conditions. The Gibson assembly technique was used to generate three plasmids harboring different numbers of tandem repeats of the *nirB* promoter (Table 1). The *phaCAB* gene cluster was activated by transcription initiation of the anaerobic promoters P_nirB_, P_2nirB_, and P_3nirB_ to regulate PHB expression under anaerobic conditions (Figure 1). Comparison of the cascaded *nirB* promoters with the native promoter is shown in Figure 3. Differences in the sequence repeat may explain the likely variation in the ability of the promoters to induce downstream gene expression. Based on previous studies, PHB production is related to the transcription level of *phaC* [33]. The mRNA level of *phaC* was analyzed by RT-PCR as an indicator of *phaCAB* operon expression intensity. In recombinant *E. coli* with plasmid pBHR-P_3nirB_, the mRNA level of *phaC* was fivefold higher than that of the strain harboring the native promoter (Figure 3A). After cultivation for 24 h in anaerobic conditions, the efficacy of gene expression was improved with the increase in the repeat number of the promoter P_nirB_. In *E. coli* S17-1 harboring pBHR-P_3nirB_, the mRNA transcription level of *phaC* was the highest, indicating that this strain may produce the highest content of PHB under anaerobic conditions (Figure 3A). Under the control of promoter P_3nirB_, the transcription level of *phaC* was significantly increased (*p* < 0.001).

Consequently, pBHR-P_3nirB_ was selected to test cell growth and PHB accumulation in the recombinant *E. coli* S17-1 under anaerobic conditions. Under the control of the native promoter, the CDW of *E. coli* S17-1 exceeded 1.23 g/L, and the PHB content in the cells was 51%. The *E. coli* S17-1 (pBHR-P_nirB_) strain accumulated 1.52 g/L CDW and contained over 41% PHB after cultivation for 48 h. In contrast, *E. coli* S17-1 (pBHR-P_2nirB_) grew to 1.63 g/L CDW and contained 65% PHB. Recombinant *E. coli* S17-1 harboring the plasmid pBHR-P_3nirB_ (three tandem repeats of *P_nirB_*) grew to approximately 1.77 g/L CDW and contained 77% PHB (Figure 3B). Under the control of *P_3nirB_*, the recombinant strain yielded the highest quantity of PHB after 48 h in shake flasks. Thus, the newly constructed promoter P_3nirB_ was used in further experiments to control the expression of the *phaCAB* operon in anaerobic conditions.

### 2.3. Disruption of the Mixed Acid Fermentation Pathway

During the past few decades, microbial production of PHB under oxygen-limited conditions has attracted considerable attention because of the reduction in fermentation costs. Under anaerobic conditions, *E. coli* undertakes mixed acid fermentation via a pathway that produces lactate, succinate, acetate, formate, and ethanol, which decrease the carbon flux into PHB accumulation (Figure 1). It was previously reported that disruption of the mixed acid fermentation pathway could increase PHB accumulation in *E. coli* [30]. To investigate the possibility of a synergistic effect, a series of mixed acid fermentation mutants—*E. coli* SA, *E. coli* SAP, and *E. coli* SAPL—were constructed (Table 1). Genes *ackA*-*pta* and *poxB*, related to the formation of acetate, and the gene *ldhA*, related to the formation of D-lactate from pyruvate, were knocked out in *E. coli* S17-1, to reduce the formation of byproducts.

Initially, the resulting recombinant strains were cultivated in the MM medium supplemented with 20 g/L glucose to produce PHB. After cultivation for 48 h, *E. coli* SA harboring plasmid pBHR-P_3nirB_ reached a yield of 77% PHB. *E. coli* SAP (pBHR-P_3nirB_) grew better under anaerobic conditions, producing over 1.34 g/L CDW that contained 84% PHB, significantly higher than the 72% PHB produced by *E. coli* S17-1 (pBHR-P_3nirB_). PHB accumulated up to 87% of 1.90 g/L CDW in the *E. coli* SAPL *ackA*-*pta*, *poxB*, and *ldhA* deletion mutant (Table 2). *E. coli* SAPL harboring the plasmid pBHR-P_3nirB_ showed the highest PHB content after cultivation compared with the wild-type and other mutant strains.

Next, the plasmid pMCS-*sacC* encoding β-fructofuranosidase was inserted into *E. coli* SAPL (pBHR-P_3nirB_). The resulting recombinant strain was then cultivated in the MM medium, with added 20 g/L sucrose as the carbon source, at 37 °C for 48 h; ampicillin and kanamycin were added to the medium to maintain the stability of the plasmids pBHR-P_3nirB_ and pMCS-*sacC*. The sucrose was rapidly hydrolyzed, and the concentrations of glucose and fructose were observed during the cultivation. PHB accumulated up to 79%, and CDW reached 7.71 g/L in *E. coli* SAPL in MM medium sucrose in shake flasks (Table 3). These results indicated that *E. coli* SAPL (pBHR-P_3nirB_, pMCS-*sacC*) was more suitable and efficient for PHB production from sucrose as the carbon source under anaerobic culture conditions.

### 2.4. Production of PHB from Sucrose in Batch Fermentation

Since the recombinant *E. coli* SAPL strain harboring the plasmids pBHR-P_3nirB_ and pMCS-*sacC* supported the highest production of PHB in flask cultures, it was used as the host strain for batch fermentation. *E. coli* SAPL (pBHR-P_3nirB_, pMCS-*sacC*) was cultured in a 5 L fermenter containing MM medium with 20 g/L sucrose. The sucrose was completely hydrolyzed into glucose and fructose, which could be efficiently used for cell growth as well as PHB production, similar to the flask cultures (see Appendix A). The host strain grew well by utilizing sucrose as a carbon source and resulting in a final CDW of 7.71 g/L. After cultivation for 48 h under anaerobic conditions, 81% of PHB was accumulated and the final concentration of PHB at the end of batch fermentation was 6.21 g/L (Figure 4). Cells of the fermentation strains were observed by SEM/TEM and were found to contain PHB inclusions (Figure 4), indicating that the sucrose pathway had been successfully constructed and *P_3nirB_* was suitable to control PHB expression under a low level of oxygen. Further experiments on the optimization of fermentation conditions to enhance PHA synthesis may be beneficial for the commercial production of PHB.

## 3. Materials and Methods

### 3.1. Bacterial Strains and Plasmids

All bacterial strains and plasmids used in this study are listed in Table 1. Three *E. coli* strains, S17-1, DH5α, and Top10, were selected as the host strains based on previous studies [21,33], to test their ability in PHB production from sucrose. *E. coli* is commonly used for genetic recombination, and the genes *ackA-pta*, *poxB*, and *ldhA* related to mixed acid fermentation were knocked out by the one-step disruption method [38]. The homologous sequence (39 bp) upstream and downstream of each target gene and the kanamycin resistance gene (*Kan*) as a selection marker flanked by FLP recognition target sites were amplified by PCR to replace the target fragment on the chromosome. The *Kan* gene was then eliminated with the help of plasmid pCP20 expressing the FLP recombinase. The resulting deletions were confirmed by DNA sequencing. Three mutant strains (*E. coli* SA, *E. coli* SAP, and *E. coli* SAPL) with one gene knockout individually or all three genes knocked out together were constructed in this study (Table 1).

Genes encoding PHB synthesis operon *phaCAB* from *Ralstonia eutropha* and *SacC* from *M. succiniciproducens* MBEL55E were cloned into plasmids pBHR68 and pBBR1MCS-2, respectively. The construction of plasmids included vector isolation, DNA amplification of genes, restriction enzyme digestion, and other molecular cloning procedures. The anaerobic promoter P*_nirB_* was cloned from the genomic DNA of *E. coli* extracted by the QIAamp DNA Mini Kit (Qiagen, Hilden, Germany). The PCR program comprised an initial denaturation at 94 °C for 3 min, followed by 30 cycles of denaturation at 94 °C for 30 s, annealing at 55 °C for 30 s, and extension at 72 °C for 2 min. The amplified sequence was tested for purity and size by 1% agarose gel electrophoresis. Genomic DNA was quantified with a NanoDrop 2000 spectrophotometer and used as the template to finish the promoter PCR. Cascaded promoter P*_nirB_* was obtained by the Gibson assembly technique.

### 3.2. Culture Medium and Growth Conditions

Luria–Bertani (LB) medium (pH 7.2), used for DNA manipulations and plasmid construction, contained (per liter) 10 g tryptone, 5 g yeast extract, and 5 g NaCl. Strains were cultured in LB medium at pH 7.2 and 37 °C. When antibiotic selection was required during cultivation, ampicillin (100 μg/mL), kanamycin (50 μg/mL), or chloramphenicol (30 μg/mL) were added to the medium depending on the plasmids.

The MM medium was used for PHB production in flask cultures and batch fermentations. This medium (pH 6.8) contained (per liter) (NH_4_)_2_SO_4_ 2.0 g, MgSO_4_ 0.2 g, Na_2_HPO_4_·12H_2_O 9.65 g, KH_2_PO_4_ 1.5 g, trace metal solution I 10 mL, and trace metal solution II 1 mL [39]. Trace metal solution contained FeSO_4_, CaCl_2_, ZnSO_4_·7H_2_O, MnSO_4_·4H_2_O, CuSO_4_·5H_2_O, and (NH_4_)_6_Mo_7_O_24_·4H_2_O and was sterilized separately. For PHB production, glucose (20 g/L) or sucrose (20 g/L) was added into the medium as the carbon source. Glucose was sterilized at 115 °C, and sucrose was sterilized using a 0.22 μm filter.

Seed cultures for flask fermentation were first prepared in LB medium incubated at 37 °C overnight in a rotary shaker at 200 rpm, and 1 mL of this overnight culture was then used to inoculate 100 mL MM medium. Aerobic growth was conducted using a 500 mL shake flask without baffles containing 100 mL culture medium under vigorous agitation at 50 rpm (Shanghai, China). Batch culture of the recombinant *E. coli* strains was performed at 37 °C in a 5 L fermenter (New Brunswick BioFlo310, MA, USA) containing 2 L MM medium plus 20 g/L sucrose. Anaerobic conditions were measured by dissolved oxygen electrodes during fermentation.

### 3.3. Real-Time Quantitative PCR

To study the induction effects of different promoters, calibration of the real-time quantitative PCR (RT-PCR) was performed based on a previously reported method [40]. Total RNA of *E. coli* was isolated, and cDNA was synthesized using an RNAprep pure Cell/Bacteria Kit and a FastQuant RT Kit. The RT-PCR assay for mRNA was performed with SuperReal PreMix (SYBR Green), using 16S rRNA (primers 16S-F: CACACTGGAACTGAGACAC and 16S-R: CTTCTTCTGCGGGTAACG) as the inner standard.

### 3.4. Electron Microscopy

PHB particles were observed by scanning electron microscopy (SEM) and transmission electron microscopy (TEM) after fermentation. For SEM, *E. coli* cells were collected and fixed with 2.5% (*v/v*) glutaraldehyde, then washed with 0.1 M phosphate-buffered saline (PBS). Next, 50%, 70%, 80%, 90%, and 100% ethanol were used to wash the fixed samples, and the samples were then further dehydrated using 2-methylpropan-2-ol mixed with ethanol (1:1). After treatment with pure 2-methylpropan-2-ol, the prepared sample was lyophilized and examined by SEM. For TEM, *E. coli* cells were pelleted by centrifugation, washed twice with PBS, and then mixed with a pre-cooled fixative solution at 4 °C overnight. After rinsing with 0.1 M phosphate buffer (pH 7.0), the samples were fixed with a 1% osmium acid solution and dehydrated with graded ethanol solutions and acetone. The prepared sample was sliced using a Leica Ultracut S ultramicrotome and was visualized using a Talos F200X transmission electron microscope (Thermo Fisher Scientific, Waltham, MA, USA).

### 3.5. Analytical Procedures

Polymers were purified from the recombinant *E. coli* strains that were cultivated in MM supplemented with 20 g/L sucrose. Briefly, cells were harvested by centrifugation at 10,000× *g* and 4 °C for 10 min and washed twice with distilled water. Cell dry weight (CDW) was measured after the cell pellets were lyophilized for more than 12 h. Methanolysis was performed by adding 2 mL CH_3_OH/H_2_SO_4_ with benzoic acid and 2 mL CHCl_3_ to the samples and incubating at 100 °C for 4 h. After cooling to room temperature and adding 1 mL ddH_2_O for stratification, the organic phase containing methyl-3-hydroxyalkanoates was analyzed by gas chromatography (GC) using a Spectra System P2000 (Thermo Separation, MA, USA) to measure the PHB content. GC was conducted using an HP-5 column (30 m, 0.25 mm) and a flame-ionization detector. Pure PHB was used as a standard sample for the GC based on a previous study [33].

To study the concentrations of sucrose, glucose, and fructose during batch fermentation, the culture supernatant was filtered through a 0.2 µm syringe filter and then analyzed by high-performance liquid chromatography (HPLC; 1260 infinity series; Agilent Technologies, CA, USA) equipped with RID Detectors (G1362A, Agilent Technologies, CA, USA) and an Aminex HPX-87H ion-exclusion column operating at 37 °C. Sulfuric acid at a concentration of 0.005 N was used at flow rate of 1 mL/min as the mobile phase.

## 4. Conclusions

Engineering microbial host strains to produce PHAs in a more efficient and cheaper way is important for their application as an environmentally friendly alternative to traditional plastics. Considering the substrate and dependence of oxygen, development of recombinant *E. coli* strains to produce PHB from sucrose as the sole carbon source under anaerobic condition can reduce both source material costs and energy consumption [41,42,43]. All the substantial components, including cost for the raw materials, utilities, and waste disposal, should be taken into account while estimating the cost [44,45]. The cost of the carbon source was mainly related with the price of different feedstocks, which have been shown to account for approximately 40% of the total PHB production costs [45]. The use of cheap carbon sources such as sucrose instead of glucose contributes to the economic feasibility of the method. On the other hand, aeration during fermentation is a major contributing factor to the fermentation costs, and the energy consumed on aeration could be reduced by using the anaerobic promoters that drive the expression of key biosynthetic genes under anaerobic conditions [33,45]. Thus, the use of promoter P*_3nirB_* could reduce the energy spent on compressed air, and the cost on feedstocks could be reduced by the utilization of sucrose, which is much cheaper than glucose, as the carbon resource. In this study, plasmid pMCS-*sacC* was constructed to utilize sucrose through the expression of β-fructofuranosidase and the promoter P*_3nirB_* was used to control PHB synthesis under anaerobic conditions. The mixed acid fermentation pathway was knocked out in *E. coli* S17-1 to reduce the synthesis of byproducts, thus *E. coli* SAPL was obtained as the host strain.

In summary, the recombinant *E. coli* SAPL strain harboring plasmids pBHR-P*_3nirB_* and pMCS-*sacC* developed in this study could successfully utilize sucrose and produce 80% of PHB under anaerobic conditions. After cultivation for 48 h under anaerobic conditions, the amount of PHB at the end of batch fermentation reached 6.21 g/L, and the cost of substrate and energy was reduced. Findings from the study demonstrated that the sucrose utilization pathway could be constructed by expressing the *sacC* gene and that the cascaded anaerobic promoter P*_3nirB_* could enhance PHB accumulation under a low level of oxygen. Using microorganisms that are capable of sucrose utilization under anaerobic conditions will be useful for establishing a cost-effective process for the production of PHB and other value-added chemicals.

## Figures and Tables

**Figure 1 molecules-27-00294-f001:**
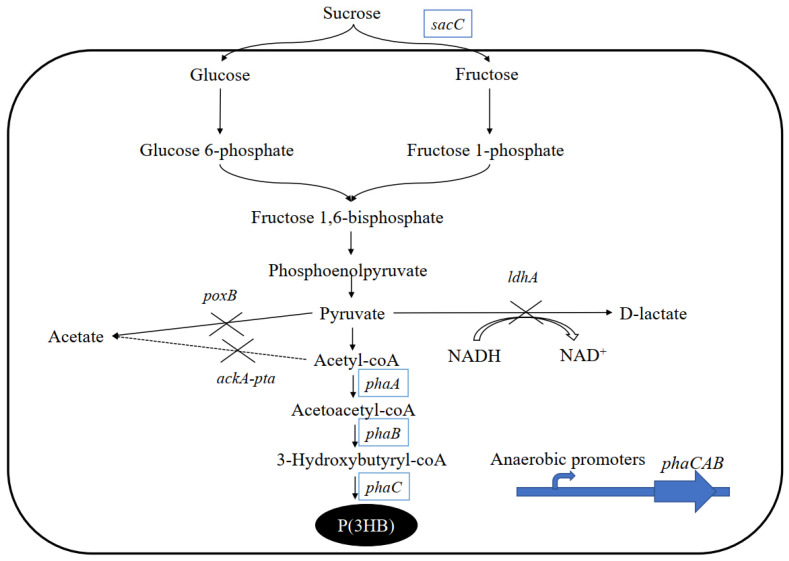
The biosynthetic pathway of PHB with flux distribution in *E. coli* using sucrose as a sole carbon source.

**Figure 2 molecules-27-00294-f002:**
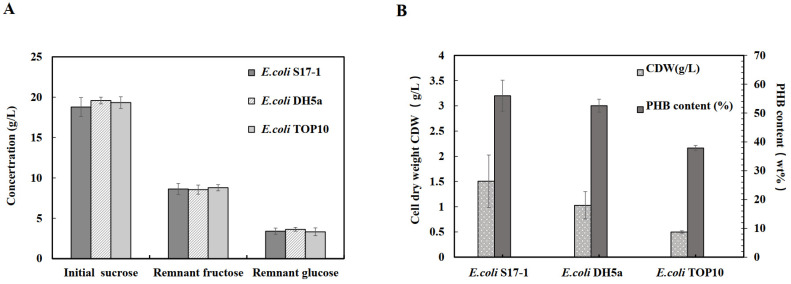
Sugar concentrations, cell dry weight (CDW), and PHB content in the culture flasks of the recombinant *E. coli* strains. (**A**) Concentrations of sucrose, glucose, and fructose were measured to investigate the utilization of sucrose; (**B**) CDW and PHB contents in the three recombinant strains of *E. coli* using sucrose as a substrate after 48 h cultivation.

**Figure 3 molecules-27-00294-f003:**
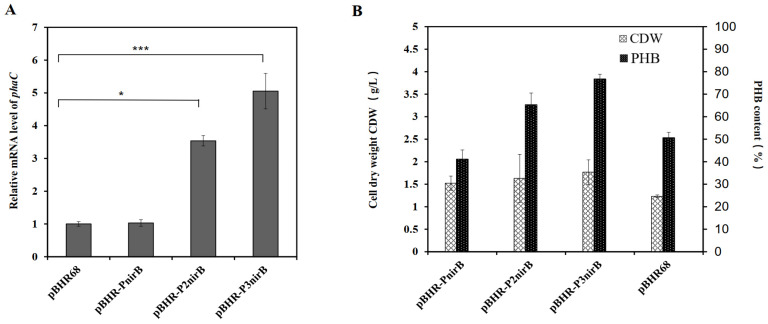
Transcription levels of *phaC* and PHB contents under different promoters. (**A**) Total RNA of four strains with different plasmids were isolated and real-time PCR assays for mRNA were performed using16S rRNA as the internal standard. Statistical analysis was performed by an unpaired two-tailed t-test. * *p* < 0.05; *** *p* <  0.001; (**B**) Cell dry weight and PHB contents of four strains with different promoters. In the X-axis of (**A**) and (**B**), plasmid names were both used to represent the strain information. pBHR-P_nirB_ represented *E. coli* S17-1 (pBHR-P_nirB_), pBHR-P2nirB represented *E. coli* S17-1 (pBHR-P_2nirB_), pBHR-P3nirB represented *E. coli* S17-1 (pBHR-P_3nirB_), and pBHR68P represented *E. coli* S17-1 (pBHR68). Data are the average of three parallel experiments.

**Figure 4 molecules-27-00294-f004:**
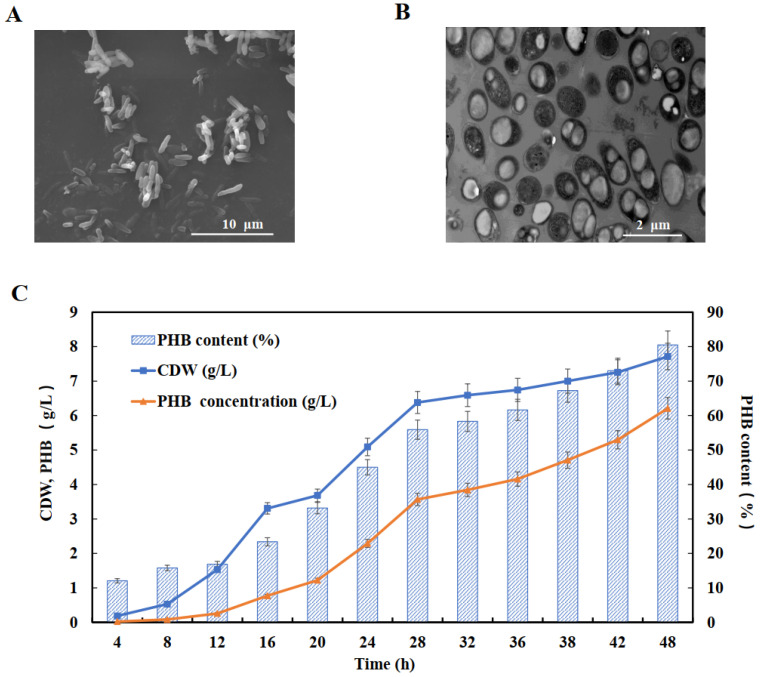
Bacterial growth and PHB production by *E. coli* SAPL harboring plasmids pBHR-P_3nirB_, pMCS-*sacC* under anaerobic conditions. (**A**) SEM micrographs of recombination strain; (**B**) TEM micrographs of recombination strain; (**C**) CDW, PHB concentration, and PHB content of recombination strain cultivated in the MM medium containing sucrose as a carbon source at 37 °C for 48 h.

**Table 1 molecules-27-00294-t001:** *Escherichia coli* strains and plasmids used in this study.

Items	Descriptions	References
Strains		
*E. coli* S17-1	*recA pro hsdR RP*4-2-*Tc::Mu-Km::Tn*7 integrated into the chromosome	[33]
*E. coli* TOP10	F-*mcrA*Δ(*mrr-hsdRMS-mcrBC*)φ80*lacZ*ΔM15Δ*lacX74 rec*A1*ara*Δ139Δ*(ara-leu)7697galUgalKrpsL(Starr)end*A1 *nupG*	Industry MTA
*E. coli* DH 5α	F-Φ80*lacZ*ΔM15Δ(*lacZYA-argF*)U169*recA1endA1 hsd*R17(rk−, mk+) *phoA sup*E44 *thi*-1 *gyr*A96 *rel*A1 *λ-*	Industry MTA
*E. coli* SA	*E. coli* S17-1 with the *ackA-pta* gene being knocked out	This study
*E. coli* SAP	*E. coli* S17-1 with the *ackA-pta* and *poxB* genes being knocked out	This study
*E. coli* SAPL	*E. coli* S17-1 with *ackA-pta, poxB* and *ldhA* genes being knocked out	This study
Plasmids		
pBHR68	A pBluescript II SK−derivative containing *phbCAB* operon from *Ralstonia eutropha* with native promoter, Amp^R^	[33]
pBHR-P*_nirB_*	*pBHR68* with the *phbCAB* operon under the control of the *nirB* anaerobic promoter	This study
pBHR-P_2*nirB*_	*pBHR68* with the *phbCAB* operon under the control of the cascaded promoter *2nirB*	This study
pBHR-P_3*nirB*_	*pBHR68* with the *phbCAB* operon under the control of the cascaded promoter *3nirB*	This study
pBBR1MCS-2	Broad-host-range plasmid, Kan^R^	Industry MTA
pMCS-sacC	pBBR1MCS-2 with the *sacC* gene from *M. succiniciproducens* MBEL55E, Kan^R^	This study

**Table 2 molecules-27-00294-t002:** Bacterial growth and PHB production by different *E. coli* strains harboring pBHR-P*_3nirB_* under an anaerobic condition.

*E. coli* Strains ^a^	CDW (g/L) ^b^	PHB Concentration (g/L) ^b^	PHB Content (wt.%) ^b^
*E. coli* S17-1 (pBHR-P_3*nirB*_)	1.52 ± 0.16	1.09 ± 0.07	72 ± 3
*E. coli* SA (pBHR-P_3*nirB*_)	1.46 ± 0.10	1.12 ± 0.12	77 ± 7
*E. coli* SAP (pBHR-P_3*nirB*_)	1.34 ± 0.11	1.13 ± 0.15	84 ± 5
*E. coli* SAPL (pBHR-P_3*nirB*_)	1.90 ± 0.47	1.65 ± 0.36	87 ± 3

^a^ Cells were cultivated in MM medium containing 20 g/L glucose under an anaerobic condition for 48 h. ^b^ Data are expressed as M ± SD, in which M refers to mean and SD to standard deviation.

**Table 3 molecules-27-00294-t003:** Bacterial growth and PHB production by *E. coli* SAPL (pBHR-P3nirB, pMCS-*sacC*) and *E. coli* S17-1 (pBHR68, pMCS-*sacC*) using sucrose as a carbon source under microaerobic conditions.

*E. coli* Strains ^a^	CDW (g/L) ^b^	PHB Concentration (g/L) ^b^	PHB Content (wt.%) ^b^
*E. coli* SAPL (pBHR-P_3nirB_, pMCS-*sacC*)	1.71 ± 0.27	1.34 ± 0.20	79 ± 6
*E. coli* S17-1 (pBHR68, pMCS-*sacC*)	1.57 ± 0.51	0.75 ± 0.19	49 ± 8

^a^ Cells were cultivated in the MM medium containing 20 g/L sucrose under an anaerobic condition for 48 h. ^b^ Data are expressed as M ± SD, in which M refers to mean and SD to standard deviation.

## Data Availability

The data presented in this study are available in the article.

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
