# Peer review of "Biosynthesis of Poly-(3-hydroxybutyrate) under the Control of an Anaerobically Induced Promoter by Recombinant Escherichia coli from Sucrose"

_molecules, 2022, doi:10.3390/molecules27010294_

Round 1

Reviewer 1 Report

This paper, entitled Biosynthesis of poly-(3-hydroxybutyrate) under the control of an anaerobically-induced promoter by recombinant Escherichia coli from sucrose, is a scholarly work and can increase knowledge on this domain. The content is relevant to Molecules and the authors provide an interesting and original study. The manuscript is well written and well related to exisiting literature. The abstract and keywords are meaningful.

I have some specific and general comments:

  • about supplementatry materials, please provide error bars in the supplementary figure 1
  • Authors wrote lines 53-54 page 2 and in Conclusions section also that "...numerous studies attempt to reduce the production cost of PHB...", what about this study? Please provide costs analysis of such method? Could you provide data about the reduction of costs ans energy? What could be the benefits or limitation in comparison with other existing methods or application? Please discuss this.
  • Please provide costs analysis and energy balance.
  • Mass balance should be given with calculation of yields.
  • How were chosen the 3 strains of E. coli? Is it from a screening in database, datacollection, strains collection, ...? or from previous studies?
  • Please provide error bars in Figure 4.
  • Is there some analysis of PHB purity or quality in comparison with other existing PHB?
  • What about the biodegradability of such PHB obtained in this study? Is there any experiments carried out about degradability?
  • Is there any experiments in continuous reactors scheduled for PHB production ?
  • Why not testing these strains on raw materials in addition of culture media in order to observe the ability of such strains to convert carbohydrates in PHB? In order to reduce the costs of production and in a concept of biorefinery, this solution must be favored from my point of view.
  • Hox was carried out the PHB fractionation / extraction and analysis?

As it, the manuscript is not fully acceptable for publication and requires at least to be revised with amendments and additional information. I recommend the following decision: ACCEPT AFTER MINOR REVISION.

Author Response

Dear reviewer,

Thank you very much for your kindly comments and useful suggestions. There is no doubt that these comments are valuable and very helpful for revising and improving our manuscript. In what follows, we would like to answer the questions you mentioned and give detailed account of the changes made to the original manuscript.

Below the comments of the reviewer 1 are point-by-point response. Please see the attachment.

Reviewer 2 Report

Overall, the manuscript is well written, showing an alternative for the biosynthesis of PHB polymers. 

The “original images” file contains the same files as “non-published”. What are the differences between them? Authors should provide the corrected images.

Some comments and suggestions:

Overall document - bacterial strains, plasmids, and genes must be in italic, and always written in the same manner.

Line 53. Authors should provide 2-3 references on the “numerous studies attempt to reduce the production cost of PHB”.

Line 70. “The E. coli wildtype strain cannot utilize sucrose as a carbon source directly”. You should explain why the wildtype cannot utilize sucrose as carbon source.

Line 129 – “MM medium”. Is this Minimal Medium?? Authors should write the full word followed by the abreviation or provide a list of observations.

Figure 2. Did authors perform any kind of statistical analysis on these results?

Line 169. How relevant is the second decimal place in the % of PHB?

Figure 3. Statistical analysis must be performed. Are the results statistically different?

Paragraph 196-204 – how significant are the second decimal place in the % values?

Line 207 – “20g/L” a spacing must be placed between the number and the unit

Line 280 – Authors say that glucose is sterilized at 115 ºC and sucrose is sterilized using a filter. Any type of glucose degradation is observed at this temperature? How can authors filter sucrose (solid – powder)??? Can you provide an explanation?

Author Response

Dear reviewer,

Thank you very much for your kindly comments and useful suggestions. There is no doubt that these comments are valuable and very helpful for revising and improving our manuscript. In what follows, we would like to answer the questions you mentioned and give detailed account of the changes made to the original manuscript.

Below the comments of the reviewer 2 are point-by-point response.Please see the attachment.
